# How can we generalise learning distributed representations of graphs?

## Abstract

This paper consists of two parts. Firstly, we specify a general framework for constructing systems capable of learning distributed representations of graphs in an unsupervised manner. We show how systems such as Deep Graph Kernels, Graph2Vec and Anonymous Walk Embeddings can be formulated under this framework. In the second part of this paper, we exemplify the construction of an instance of this framework by proposing a model which extends Graph2Vec, called G2DR. This model extends Graph2Vec's current implementation to enable application on unlabelled graphs and tackles issues of diagonal dominance through pruning of the substructure vocabulary. These extensions help G2DR achieve state of the art results in downstream application of learned embeddings in graph classification tasks over datasets with small labeled graphs in binary classification to multiclass classification on large unlabelled graphs using an off-the-shelf support vector machine.

## 1 Introduction

A fundamental prerequisite for machine learning algorithms to *learn* about input data is the ability to discern one observation from another. Even more powerful is the ability to determine how *similar* or dissimilar such observations are from one another to make more detailed associations. For observations represented in Euclidean space with feature vectors the concept of similarity between observations is intuitive as it may be computed as *distance* using Euclidean distance or cosine similarity formulas. Unfortunately, for observations represented as graphs defining the notion of similarity, and even more so designing methods for computing similarity has been a long ongoing challenge in maths and computer science. This is partly because assessing comparability of graphs does not only consist of looking for similar elements (nodes) but also structural similarities between the substructures within. An obvious real world example on the importance of this distinction can be found in chemistry where molecules called isomers exhibit identical chemical formulas but different structural properties which induce different behaviours and traits (Petrucci et al., 2017). The difficulty of comparing graphs is highlighted by the graph isomorphism test (Garey & Johnson, 1990), which despite its complexity only gives a binary evaluation of the structural equivalence between two graphs which is insufficient for demands of fine grained machine learning tasks.

Consequently, the graph learning domain predominantly features kernel methods which approximate the comparability of graphs using invariants or substructures such as nodes, subgraphs and random walks within the graphs (Vishwanathan et al., 2010). Whilst they are powerful and intuitive, such kernels are dependent and often tied to certain methods such as support vector machines (SVM) to perform learning tasks (Yanardag & Vishwanathan, 2015). Hence, existing methods utilizing kernel methods are often unable to handle other downstream learning tasks such as regression or clustering without significant revision.

More recently, deep learning approaches for graph representation learning have gained significant research activity with the successful interpretation of graph convolutional methods for learning node representations (Kipf & Welling, 2017; Scarselli et al., 2009; Veličković et al., 2018). Representations at the graph level are then constructed through application of different pooling operations which aggregate node representations into a single representation for the entire graph (Ying et al., 2018; Goyal & Ferrara, 2018).

Our approach takes a different perspective through the distributive modelling of graphs based on discrete substructure/subgraph patterns across the graph dataset. This builds fixed size vector representations of graphs within a distributed vector space created by training a neural language model which exploits the distributive hypothesis (Harris, 1954). The positions of the distributed vector representations are contextualized by the subgraph patterns within graphs with respect to the patterns seen within other graphs across the dataset. This is analogous to the distributive modelling of arbitrary sized text documents in Le & Mikolov (2014), but the implementation is typically closer to word2vec models in Mikolov et al. (2013). We show that the construction and application of distributed representations of graphs can be summarized in a three step framework. The first step involves reducing graphs into higher order subgraph patterns to create *graph documents* summarizing the patterns seen in each graph. Together the graph documents constitute a *corpus* of graphs. The second stage trains a neural language model on the graph document corpus, thereby building a distributed embedding for each graph within the dataset. In the third and final stage the embeddings may be used with off-the-shelf learning systems for downstream tasks such as classification, regression, clustering, or even in transfer learning scenarios.

To exemplify the construction of a system described using this framework, we propose a system which extends Graph2Vec called G2DR. Our proposed extensions allow distributed representations to be learned on labelled and unlabelled graphs by incorporating the suggestion to use node degrees for unlabelled nodes from (Shervashidze et al., 2011). Furthermore we directly use Shervashidze et al. (2011)'s Weisfeiler-Lehman (WL) node relabeling algorithm for a more computationally efficient algorithm for constructing the subtree vocabulary of the graph dataset which is equivalent to the WL algorithm described in Graph2Vec. Like Graph2Vec, G2DR uses a skipgram with negative sampling to learn distributed representations of graphs based on subtree patterns induced with the WL algorithm. However we introduce a vocabulary pruning heuristic to help the skipgram model further alleviate the issue of diagonal dominance which typically hurt graph kernel methods. This heuristic helps G2DR during our evaluation of the learned embeddings on graph classification tasks with publicly available graph kernel datasets (Kersting et al., 2016). Datasets and tasks were selected to cover a range of graphs from small labelled graphs to large unlabelled graphs exhibiting complex substructures. To be comparatively fair within our assessment with kernel methods that inspire G2DR we utilised support vector machines (SVM) on the distributed representations and outperformed kernel methods and stands competitively amongst popular supervised GNN methods. This helps validate the distributed perspective as a useful inductive bias (Battaglia et al., 2018) for constructing graph level representations alongside pooling efforts, and as a possible research avenue for more unsupervised techniques to graph-structured data.

In section 2 we provide a comprehensive background into graph learning and related methods such as graph kernels. Section 3 describes our 3 stage framework for constructing models. Section 4 describes the construction of G2DR using the framework in Section 3. Section 5 describes the downstream application of the distributed representations in graph classification tasks with details about datasets and experimental setup, followed by results and discussion.

## 2 BACKGROUND AND RELATED WORK

Many real world phenomena such as chemical compounds (Petrucci et al., 2017), protein structures (Borgwardt et al., 2005), application process calls (Gascon et al., 2013), and social networks (Yanardag & Vishwanathan, 2015) can be naturally represented using graphs. For example, in chemistry the graph makes an intuitive model for a molecule where nodes represent atoms and edges the bonds between them. Here the graph is an appropriate representation as it captures not only the presence of the atoms in the molecule, but the edges also capture the specific bonding patterns between the atoms which is important for distinguishing isomers that a classical chemical formula cannot describe (Petrucci et al., 2017). In other words, the resulting graph topology created by the relationships between the nodes in a graph reveal a structural complexity that can be analysed as a source of information in pattern recognition problems.

Described more formally, a graph is an abstract structure which defines a set of entities which are related in some way. Graphs contain *nodes* representing said entities with related nodes being connected by an *edge* which records the relation. We define $\mathcal{G} = (V, E)$ as a graph where $V$ is a set of nodes and $E \subseteq (V \times V)$ be a 2-tuple set of edges in the graph. Hence if $u$ and $v$ are nodes in $\mathcal{G}$, their

relation is recorded with an edge as $(u, v) \in E$. The *neighbours* of a node $v$ in graph $\mathcal{G} = (V, E)$, is the set of nodes which share an edge with $v$, denoted $\mathcal{N}(v) = \{u | (v, u) \in E\}$.

Graphs can be categorised depending on the attributes of the nodes and edges. A *labelled graph* is a graph whose nodes or edges have labels, which may or may not be unique. Nodes and/or edges can be labelled, with the graphs then being called node- or edge-labelled graphs respectively. Otherwise it is simply known as an *unlabelled graph*. Edges can either directed or undirected. Directed edges are uni-directional relations from a starting to node $u$ to a target node $v$ and recorded as $(u, v) \in E$ and $(u, v) \neq (v, u)$. Undirected edges describe bi-directional relationships between nodes $u$ and $v$, hence $(u, v) = (v, u)$.

## 2.1 GRAPH LEARNING AND KERNEL METHODS

An operational assumption made in learning with graph-structured data is that similar phenomena represented by graphs will also exhibit similar topological properties. Hence the ability to quantify the similarity of graph topologies is central to graph learning algorithms. As topological patterns are not intuitively well represented using classic feature vectors, research has predominantly focused on using *kernel methods* for machine learning tasks involving graphs. Kernel methods are machine learning algorithms which rely on a *kernel* for the pattern recognition task. Kernels are functions which define a relation or more contextually, a similarity over pairs of data points using their raw representations. Subsequently one can use kernel methods which can operate on kernels such as support vector machines (SVMs) (Yanardag & Vishwanathan, 2015).

Ideally a kernel would be a similarity function $\text{sim}(\mathcal{G}, \mathcal{G}') = d, d \in \mathbb{R}^+$ where $d$ or "distance" between graphs $\mathcal{G}$ and $\mathcal{G}'$ is small if they have similar structural properties, and a larger distance otherwise. The most intuitive measure of similarity is the binary indication of whether two graphs are topologically identical or *isomorphic*.

This is also known as the Graph Isomorphism (GI) test. Despite being a rudimentary measure of similarity, the complexity of the GI test is in NP and has neither been proven to be NP complete nor solved by a polynomial time algorithm (Garey & Johnson, 1990). Out of the twelve computational complexity problems listed in Garey and Johnson (Garey & Johnson, 1990), only the GI problem and integer factorisation remain unsolved. In addition to being computationally expensive, the binary measure of similarity provided by GI based measures requires graphs to be identical or contain large identical subgraphs in order to be considered similar. This is too restrictive to be used effectively by machine learning methods. As a result a number of more flexible kernels based on approximate and inexact matching of graphs were proposed to address this problem.

Examples of these approximate kernels include *graph edit distance* methods and *invariant based* methods. Graph edit distance methods as proposed by Bunke & Allermann (1983); Neuhaus & Bunke (2005) define a set of graph edit operations and associates a "cost" with each operation. The distance between the graphs can then be approximated by the minimum number and cost of edits needed to transform one graph into another. Slightly less intuitive, but powerful kernels exploit graph invariants. Kondor & Borgwardt (2008) introduced the skew spectrum where the invariant feature known as the graph skew is computed from the graph and extracted bispectral invariants can be compared into a kernel value. Less than a year later, Kondor et al. (2009) proposed the graphlet spectrum which computes a spectrum of matrices relative to a set of subgraphs. These features capture the number and position of the subgraphs which could then be compared between graphs.

Yanardag & Vishwanathan (2015) noted that kernel methods such as the graphlet spectrum are part a larger family of graph kernels which evaluate the similarity between graphs $\mathcal{G}$ and $\mathcal{G}'$ by decomposing them into atomic substructures such as random walks, shortest paths, graphlets, and subgraph patterns. The kernel value is then calculated by some function such as counting the number of common substructures over $\mathcal{G}$ and $\mathcal{G}'$. These kernel values would then be exploited by kernel methods performing the machine learning task. Such count based graph kernels can largely be grouped into three major families: those based on finite size subgraphs (Kondor et al., 2009; Horváth et al., 2004; Shervashidze et al., 2009), subtree patterns (Shervashidze et al., 2011; Shervashidze & Borgwardt, 2009; Ramon & Gärtner, 2003), and walks or paths (Borgwardt & Kriegel, 2005; Kashima et al., 2003; Vishwanathan et al., 2010).

Graph kernels are intuitive, efficient and perform well on smaller benchmark datasets however exhibit two limitations. Firstly, most kernels do not create explicit graph embeddings. This makes many out of the box machine learning algorithms that rely on vector embeddings such as Random Forests, Neural Networks, Naive Bayes, etc. unable to work with graph data. Secondly, the substructures which the graphs are decomposed to have to be determined manually with well defined functions that help extracting such substructures from graphs. When such substructures are used in very large datasets this can lead to building extremely high-dimensional, sparse, and non-smooth representations of graphs (Narayanan et al., 2017).

## 2.2 DEEP LEARNING APPROACHES

More recently, success in the node classification tasks with graph convolutional networks (GCN) (Kipf & Welling, 2017) and successful translation of attention models (GAT) (Veličković et al., 2018) brought a flurry of attention to learning representations of substructures and whole graphs using deep learning. Whilst a majority of research in this regard has focused on learning representations of nodes (Goyal & Ferrara, 2018; Battaglia et al., 2018) methods have been proposed to pool or aggregate substructure representations into graph level representations (Goyal & Ferrara, 2018). A notable example is Ying et al. (2018) which hierarchically clusters and coarsens substructure representations towards a graph representation with stacks of graph neural networks. These have produced excellent empirical results on graph classification datasets overcoming some of the limitations of graph kernels on datasets with large graphs. The recent GIN in Xu et al. (2019) is currently the most powerful supervised example and is included in this work for comparison. However these methods face interesting challenges as they are sensitive to network initialisations and jumping knowledge structures, and simple structure-unaware MLPs and single layer GCN models have been shown to display comparable performance to the most powerful models despite their great algorithmic and memory complexity as shown in Luzhnica et al. (2019).

One deep learning approach that works slightly differently is Niepert et al. (2016) PATCHY-SAN. This method incorporates ideas from work in kernels and assigns labels to nodes using the labelling procedure from the Weisfeiler-Lehman kernel (Shervashidze et al., 2011), and sorts the node labels into a line. Subsequently, PATCHY-SAN defines a receptive field around each node by selecting a fixed number of nodes in $d$-hop neighbourhood where $d$ is a natural number chosen by the practitioner. It then uses a standard convolutional neural network to learn a representation of the graph. This method features prominently as a state of the art supervised approach to graph classification.

## 2.3 DEEP GRAPH KERNELS AND GRAPH2VEC

Our work is inspired by Deep Graph Kernels (Yanardag & Vishwanathan, 2015) and Graph2Vec (Narayanan et al., 2017). Yanardag & Vishwanathan (2015) recognised that many graph kernel methods can be formulated as instances of the R-Convolutional kernel framework (Haussler, 1999) which decomposes discrete structures such as graphs into smaller substructure patterns to define kernels compatible with kernel methods like SVMs. Yanardag & Vishwanathan (2015) then utilised graph-edit distance methods or learning methods based on neural language models to compute a similarity matrix over the substructures to define kernel functions.

Graph2Vec (Narayanan et al., 2017) is a specific model which builds upon principles of Deep Graph Kernels. The authors defined a recursive subgraph decomposition algorithm based on the Weisfeiler-Lehman test (Weisfeiler & Lehman, 1968) to find subtree patterns for each node in labelled and undirected graphs, subtree patterns are recorded as strings in documents for each graph. An extension of this specific algorithm based on Shervashidze et al. (2011) node relabel algorithm is presented in the Section 4. A key deviation from Deep Graph Kernels is the subsequent input of the documents into a neural language model based on word2vec's skipgram architecture (Mikolov et al., 2013). This produced a scalable unsupervised approach to building distributed vector representations of graphs which captures generic structural properties of graphs. Yet the effectiveness of the embeddings, at least where data is publicly available, was only shown with classification datasets consisting of small labelled graphs. Nonetheless, the potential of the underlying methodology and potential motivates the general framework and the extensions we propose through G2DR described in the Section 4.

# 3 FRAMEWORK FOR CREATING MODELS WHICH LEARN DISTRIBUTED REPRESENTATIONS OF GRAPHS

We present a 3 stage framework for constructing models that can learn distributed representations of graphs.

1. **Substructure pattern extraction**: The first stage consists of decomposing each graph into subgraph patterns using methods that respect the R-Convolution kernel framework, such as graphlets, subtree patterns, and walks. During this process a hash function is used to record patterns to unique string labels within a dictionary which defines a *vocabulary* of the different subgraph patterns found across the dataset. The string labels of patterns found in a graph are recorded in an associated text document called a *graph document*. This process creates a collection of graph documents corresponding to each graph in the dataset forming a graph document *corpus* representative of the input graph dataset.

2. **Distributed representation learning**: This stage borrows concepts from statistical language modelling where the distributional hypothesis (Harris, 1954) is used to imbue *semantic meaning* to words and documents based on *context*. This distributional hypothesis suggests that words which are used and exist in the same context have similar meanings (Harris, 1954). We may adopt the same view about graphs by hypothesising that graphs which exhibit similar subgraph patterns entail similar properties and classifications. Therefore fixed distributed representations of graphs may be learned via neural language models such as word2vec (Mikolov et al., 2013), doc2vec (Le & Mikolov, 2014), or GloVe (Pennington et al., 2014) on the corpus of graphs of stage 1.

3. **Downstream application:** The third and final stage involves the downstream application of the distributed representations learned in stage 2. As the learning models in stage 2 produce fixed-size vector representations of each graph, any off-the-shelf learning algorithm can readily be applied such as SVMs and multi-layer perceptrons for classification, or K-Means for clustering, or decision trees for regression.

Specific systems such as Deep Graph Kernels, Graph2Vec, and Anonymous Walk Embeddings can all be described under this framework. Deep Graph Kernels (Yanardag & Vishwanathan, 2015) induce graphlets, shortest-paths, subtree patterns within the first stage associating the subgraph patterns with graphs to define a context. Within the second stage Deep Graph Kernels use a word2vec model where a graph is contextualised by its induced subgraphs. Graph2Vec (Narayanan et al., 2017) uses subtree-patterns extracted using an algorithm based on the Weisfeiler-Lehman graph isomorphism test and contextualises each graph with its induced subtrees in the first stage. In the second stage, Graph2Vec uses a skipgram with negative sampling on the graph-induced substructure pairs. The more recently published Anonymous Walk Embeddings (AWE) (Ivanov & Burnaev, 2018) is similar to Graph2Vec except it induces anonymous walks from each graph for the first step, and uses these to build distributed representations with a skipgram model. In the next section, we present G2DR a model which extends Graph2vec described using this framework above.

# 4 PRESENTING G2DR IN TERMS OF THE FRAMEWORK

## 4.1 STAGE 1: SUBTREE DECOMPOSITION VIA WEISFEILER-LEHMAN REDUCTION

In the first stage G2DR utilises a subtree decomposition algorithm based on the Weisfeiler-Lehman (WL) graph isomorphism test. Amongst the possible algorithms tackling graph isomorphism the Weisfeiler-Lehman (WL) test (Weisfeiler & Lehman, 1968) stands out as a particularly effective solution that differentiates graphs based on the composition of the subtree patterns extracted from the nodes, and works for most cases except those covered in Cai et al. (1992). In its essence, the WL test consists of iteratively assigning unique labels to the multisets formed by a nodes label with the labels of its neighbours. As this is performed on each node within the graph, the $d^{\text{th}}$ iteration of node relabeling is equivalent to encoding a subtree pattern formed of a node and the neighbourhood within $d$-hops. If the composition of assigned labels differs between graphs at any iteration, the WL test considers them non-isomorphic.

We wish to find subtree patterns of degree/depth $d$ for every node within the graph of every graph in the dataset. This is achieved as a byproduct of the WL test's node relabeling (Shervashidze et al., 2011) and is fully described in Algorithm 1 for reference. G2DR utilises subtree patterns over other atomic substructures which could compose graphs for two reasons. Firstly, subtrees are higher order substructures over nodes which offer richer description of not only the nodes in a graph but the local neighbourhood structures which when sampled can give better embeddings of graphs. Secondly, compared to linear patterns such as walks and paths, non-linear subtree patterns capture the structure of node neighbourhoods more generally (Narayanan et al., 2017).

---

**Algorithm 1:** WL-Relabel $(\mathbb{G}, d)$

---

> **Input**   : $\mathbb{G} = \{\mathcal{G}_1, \mathcal{G}_2, ..., \mathcal{G}_n\}$ a set of $n$ graphs
> $\quad\quad\quad\quad$ $d$ the desired degree of rooted subgraph found from each node
> **Output:** $\mathbb{G}$ with each of the node relabellings made at each iteration saved in all nodes of all
> $\quad\quad\quad\quad$ graphs

1 **for** $i = 0$ **to** $d$, **by** 1 **do**

2 $\quad$ Multiset-Label determination
> - if $i = 0$, set $M_i(v) = l_0(v) = \lambda(v)$
> - if $i > 0$, assign a multiset-label $M_i(v)$ to each node $v$ in $\mathcal{G}$ for all $\mathcal{G} \in \mathbb{G}$ which consists of the multiset $\{l_{i-1}(u) | u \in N(v)\}$

3 $\quad$ Sorting each multiset
> - Sort elements in $M_i(v)$ in ascending order and concatenate them into a string $s_i(v)$
> - Concatenate $l_{i-1}(v)$ as a prefix to $s_i(v)$ and set the resulting string as $s_i(v)$

4 $\quad$ Label Compression
> - Sort all of the strings $s_i(v)$ for all $v$ form $\mathcal{G} \in \mathbb{G}$ in ascending order.
> - Map each string $s_i(v)$ to a new compressed label, using a hash function $f : \Sigma^* \rightarrow \Sigma$ such that $f(s_i(v)) = f(s_i(w))$ iff $s_i(v) == s_i(w)$

5 $\quad$ Node Relabelling
> - Set $l_i(v) = f(s_i(v))$ for all nodes in $\mathcal{G} \in \mathbb{G}$

6 **end**

---

Algorithm 1, generates rooted subgraphs (subtrees) $t^{(d)}$ of depth $d$ around every node $v$ of graph $\mathcal{G} \in \mathbb{G}$. In this iterative algorithm starting from $i = 0$ to $i = d$, a hash function is initialised at the beginning of each iteration which maps the subgraph pattern to compressed multiset labels. When $i = 0$ no subgraph needs to be extracted and hence the original label of node $v$ is returned (line 1). If the graph is unlabelled, the degree of a node is set as its label. Using degree as the initial node label can also apply to labelled graphs and will effectively construct a vocabulary of structural patterns (Shervashidze et al., 2011). When $i > 0$ the labels of the previous iteration are used to create new compressed labels. For each node $v$ we find the previous iterations' labels of all $v$'s neighbours as a multi set $\{l_{i-1}(u) | u \in \mathcal{N}(v)\}$ and sort the elements in ascending order. The sorted set is then turned into a string $s_i(v)$ with $v$'s label in the previous iteration prepended to it. These strings are passed into the hash function to assign new compressed labels for the multiset string $s_i(v)$. Note that this makes sure that if the same subgraph pattern of $v$ is found in another graph, that node is given the same compressed label due to the hashmap, simultaneously building a "vocabulary" $\mathcal{V}$ of the different subtree patterns in $\mathbb{G}$. Furthermore, at each iteration of this algorithm compressed labels $l_i(v)$ are produced which correspond to subtree patterns of depth $i$ rooted at $v$.

At the end of algorithm 1 each graph $\mathcal{G}_j \in \mathbb{G}$ will have $d$ relabellings for each of the nodes $v \in \mathcal{G}_j$ in string form $s_i(v)$ which we store as attributes inside of a NetworkX graph object (Schult, 2008). For each graph $G \in \mathbb{G}$, we generate a *graph document* by writing each of the compressed relabellings $s_i(v)$ for all $i \in [1, d]$ of every node $v \in G$ into a text file. All of the graph documents together form our corpus of graphs. Alternatively one can see this as a dataset $\mathcal{D}$ recording every pair of graph with its induced subtree patterns $(\mathcal{G}_i, t), t \in \mathcal{V}$.

Algorithm 1 deviates from the algorithm presented in the Graph2Vec paper (Narayanan et al., 2017) where multisets representing subtree patterns are recursively constructed using the labels of neighbours of a target node for each node of all graphs in the dataset, and then assigned to that node. Instead the proposed WL-Relabel algorithm iteratively assigns compressed multiset labels based on a nodes previous multiset label, and in the case of an unlabelled graph, runs an initial relabelling based on the degree of the node. Our changes bring three advantages. One, it allows extraction of subgraph patterns for unlabelled graphs, allowing G2DR to learn distributed representations of unlabelled graphs and labelled graphs; and can apply directly to directed graphs. Two, during the iterative process, the subtree patterns of each degree prior to the user specified degree are observed and can be used or saved without finding them again. Three, the compressed labels are a memory efficient alternative to saving entire subgraph patterns for each node.

## 4.2 Stage 2: Distributed representation learning with skipgram using negative sampling

Once the graph documents have been generated using algorithm 1 it is possible to learn a distributional vector space in a completely unsupervised manner using a distributed vector space model. We follow Graph2Vec's example (Narayanan et al., 2017) and use a skipgram model with negative sampling (Mikolov et al., 2013). In order to exploit the distributive hypothesis the "context" of a graph is defined as the subtree patterns induced during stage 1. The data $\mathcal{D}$ upon which the skipgram is trained consists of Graph-Subtree pairs $(\mathcal{G}_i, t) \in \mathcal{D}$ where graph $\mathcal{G}_i \in \mathbb{G}$ and $t \in \mathcal{V}$ is a subtree pattern in the vocabulary induced during application of the WL-Relabel algorithm. Distributed representations can then be learned by optimising the objective function:

$$\mathcal{L} = \sum_{\mathcal{G}_i \in \mathbb{G}} \sum_{t \in \mathcal{V}} |\{(\mathcal{G}_i, t) \in \mathcal{D}\}|(\log \sigma(\Phi_i \cdot \mathcal{S}_t) + k \cdot \mathbb{E}_{t_N \in P_D}[\log \sigma(-\Phi_i \cdot t_N)]) \quad (1)$$

Where $\Phi \in \mathbb{R}^{|\mathbb{G}| \times \delta}$ is the $\delta$ dimensional matrix of graph embeddings of the graph dataset $\mathbb{G}$, and $\Phi_i$ is embedding for $\mathcal{G}_i \in \mathbb{G}$. Similarly, $\mathcal{S} \in \mathbb{R}^{|\mathcal{V}| \times \delta}$ are the $\delta$ dimensional embeddings of the subtree patterns in the vocabulary $\mathcal{V}$ so $\mathcal{S}_t$ represents the vector embedding corresponding to subtree pattern $t$. The embeddings of the subtree patterns are also tuned but ultimately not used, as we are interested in the graph embeddings in $\Phi$. $k$ is the number of negative samples with $t_N$ being the sampled context subtree, drawn according to the empirical unigram distribution $P_D(t) = \frac{|\{t | \forall G_i \in \mathbb{G}, (G_i, t) \in \mathcal{D}\}|}{|D|}$.

### 4.2.1 Further tackling diagonal dominance with vocabulary pruning

A main limitation of graph kernel methods based on computing kernels across frequency vectors of induced subgraph patterns is the number of subgraph patterns used to compare graphs to one another. For subtree patterns extracted using the WL algorithm the degree of subtrees to be extracted from the graphs in the dataset directly affects the vocabulary size. As the subtree degree to be analysed increases, the number of unique substructure patterns which enter the vocabulary also increases. For example, using the WL relabeling algorithm on the NCI1 dataset induces 267 subtrees of degree 1, 4033 of degree 2, and 22923 unique subtree patterns of degree 3 across the graphs of NCI1.

Graph kernels would then typically have to compute the kernel between two graphs based on frequency vectors whose dimensions is the size of the vocabulary considered. Consequently as the number of features (vocabulary size) grows, we run into the sparsity problem, where only a few substructures will be common across the graphs. This leads to the phenomenon known as *diagonal dominance*, where graphs become more similar to themselves but more distant from other graphs in the dataset (Yanardag & Vishwanathan, 2015).

The main contribution of Deep Graph Kernels and subsequent methods was the use of the embedding method to produce dense low dimensional distributed representations. However, we can see from Equation 1 the graph embeddings will still be affected by the embeddings $S_t$ of infrequently appearing subgraph patterns. Hence, graph embeddings are still indirectly are affected by the concept of diagonal dominance. A naive yet natural fix which addresses this concern on infrequent subgraphs is to modify the vocabulary so that a subgraph pattern has to occur a set minimum number of times in the dataset to be included in the vocabulary and hence the skipgram model. We chose

to set a minimum threshold of 2 occurrences to be included in the vocabulary if half the number of singleton subgraph patterns was higher than the number of graphs in $\mathbb{G}$.

## 5 STAGE 3: EVALUATION ON DOWNSTREAM GRAPH CLASSIFICATION

After training the graph neural language model, $\Phi$ contains the distributed representations of every graph $\mathcal{G}_i \in \mathbb{G}$, where the embedding of $\mathcal{G}_i$ is denoted $\Phi_i$. The task agnostic nature of the embeddings allows any downstream task such as classification, regression, clustering, etc. to be performed on the graphs of $\mathbb{G}$ simply by applying the appropriate methods which accept vector inputs.

We utilised the embeddings trained on graph classification datasets to assess the suitability of the approach, and as a benchmark against other graph learning methods. One of the key motivations of this work was to assess the suitability of the distributed representations beyond small labeled graphs towards larger (also unlabelled) graphs where the limitations of specification in graph kernels were expected to show. For ease of replicability a suite of public datasets were selected from Kersting et al. (2016) based on frequency in papers and size. The selection of benchmark datasets and tasks range from smaller, classic datasets with graphs averaging 17 nodes in size tasked on binary classification, to significantly larger graphs averaging over 500 nodes in size over 11929 examples in a multi-community prediction task. This list includes the MUTAG (Debnath et al., 1991) , PTC (Helma et al., 2001), PROTEINS (Borgwardt et al., 2005) , NCI1, NCI109 (Wale et al., 2008), REDDIT-BINARY, REDDIT-MULTI-5K, REDDIT-MULTI-12K (Yanardag & Vishwanathan, 2015) datasets. The datasets represent applications from a number of research domains including: chemoinformatics, bioinformatics, and social networks whose properties are summarised in Table 3 in the supplementary material.

We were particularly keen to investigate the case of unlabelled graphs and large datasets with multiple graph labels where the 'large' applies to both the number of data points as well as the size of the graphs therein. The large graphs make the representation learning inherently more complex as the number of expected subgraph patterns substantially increases (Yanardag & Vishwanathan, 2015) and questions the necessary specificity required to compare large graphs to each other as well as large graphs to small graphs.

### 5.1 EXPERIMENTAL SETUP

For fairer comparative analysis with graph kernel methods and experimental setups of Deep Graph Kernels Yanardag & Vishwanathan (2015) and Graph2Vec Narayanan et al. (2017), G2DR uses an off-the-shelf $C$-Support Vector Machine implemented in the SciKit-Learn package (Pedregosa et al., 2011) on the learned embeddings for each dataset. 10 fold cross validation was used to evaluate test classification accuracy. The $C$ value for each fold was independently tuned using training data from that fold. In order to exclude random effects of the fold assignments, each experiment was repeated 10 times, and mean classification accuracies with their standard deviations were recorded.

Results are presented over two section, one dedicated to datasets whose graphs are labelled, and the other dedicated to datasets of graphs which are initially unlabelled and hence get labelled through Algorithm 1 using the degree of the nodes as in Shervashidze et al. (2011). For comparative purposes of existing methods we have selected prominent methods applicable to each of the graph types and recorded the results they have published.

G2DR was experimented using embedding dimensions of 16, 32, 64, 128, 256, 512, 1024, 2048 across a consideration of subtree degrees of 1,2,3. For each configuration the neural network was trained over 1000 epochs with an initial learning rate of 0.5 set on an linear decay over the epochs towards 0.001. A negative sampling size of 10 was used throughout as in Graph2Vec (Narayanan et al., 2017). The best downstream results on the different embedding configurations is reported in the results of the next section.

### 5.2 RESULTS: LABELLED GRAPHS

Table 3 shows that the labelled graph datasets have a range of different properties, including datasets with small graphs and a small number of distinct labels such as the MUTAG dataset to larger datasets with larger graphs and number of distinct labels. It was expected that lower-order substructure

Table 1: Mean classification accuracy and standard deviation on labelled graph classification, italicised results indicate an unsupervised model for learning representations or kernels.

| Dataset | MUTAG | PTC | PROTEINS | NCI1 | NCI109 |
|---|---|---|---|---|---|
| node2vec | *72.62 ± 10.20* | *55.85 ± 8.00* | *57.49 ± 3.57* | *54.89 ± 1.61* | *52.68 ± 1.56* |
| sub2vec | *61.05 ± 15.79* | *59.99 ± 6.38* | *53.03 ± 5.55* | *52.84 ± 1.47* | *50.67 ± 1.50* |
| Graphlet Kernel | *81.66 ± 2.11* | *57.26 ± 1.41* | *71.67 ± 0.55* | *62.28 ± 0.29* | *62.60 ± 0.19* |
| Shortest Path Kernel | *85.22 ± 2.43* | *58.24 ± 2.44* | *75.07 ± 0.54* | *73.00 ± 0.24* | *73.00 ± 0.26* |
| WL Kernel | *80.72 ± 3.00* | *56.97 ± 2.01* | *72.92 ± 0.56* | *80.13 ± 0.50* | *80.22 ± 0.34* |
| Deep GK | *82.66 ± 1.45* | *57.32 ± 1.13* | *71.68 ± 0.50* | *62.48 ± 0.25* | *62.69 ± 0.23* |
| Deep SP | *87.44 ± 2.27* | *60.08 ± 2.55* | *75.68 ± 0.54* | *73.55 ± 0.51* | *73.26 ± 0.26* |
| Deep WL | *82.94 ± 2.68* | *59.17 ± 1.56* | *73.30 ± 0.82* | *80.31 ± 0.46* | *80.32 ± 0.33* |
| Graph2Vec | *83.15 ± 9.25* | *60.17 ± 6.86* | *73.30 ± 2.05* | *73.22 ± 1.81* | *74.26 ± 1.47* |
| AWE | *87.87 ± 9.76* | - | - | - | - |
| PATCHY-SAN (K=10) | 88.95 ± 4.67 | 62.29 ± 5.68 | 75.89 ± 2.76 | 76.34 ± 1.68 | 76.37 ± 1.43 |
| SUM-MLP (GIN-0)) | **89.4 ± 5.60** | **64.60 ± 7.00** | **76.20 ± 2.80** | 82.7 ± 1.70 | - |
| SUM-MLP (GIN-$\epsilon$) | **89.0 ± 6.00** | 63.70 ± 8.20 | 75.9 ± 3.80 | 82.7 ± 1.60 | - |
| G2DR + SVM | *89.12 ± 1.71* | *63.84 ± 3.19* | *75.61 ± 1.31* | ***83.70 ± 0.70*** | ***83.79 ± 0.57*** |

embedding methods such as node2vec (Grover & Leskovec, 2016) and sub2vec (Adhikari et al., 2017) would perform well in datasets of small graphs with few labels, but gradually perform worse as the graphs became more complex. As shown in Table 1, these methods initially perform well on smaller datasets, but suffer from the naive agglomeration schemes for larger datasets, barely performing better than random on NCI1 and NCI109. Sub2vec samples only one random walk of fixed length from the given graph and learns its representations using fixed length linear context skipgram models. This prevents sub2vec from learning structurally meaningful embeddings of the entire graph, and suffers considerably when applied to larger graphs as can be seen in the difference between MUTAG and NCI109.

The graph kernels perform better overall, as they have been purpose built for this task, with the WL-kernel performing particularly well in the larger datasets, which also initially prompted the decision to use subtrees in G2DR. The deep graph kernels, Graph2Vec and AWE, attempt to overcome the limitation of previous methods by addressing the sparsity problem of graph kernels. This raises the performance significantly over node2vec and sub2vec, however only marginally better than the traditional graph kernels upon which they are based.

The proposed embedding learning system combined with an out-of-the-box C-SVM returns highly competitive results. The success of this method is attributed to the descriptive power of non-linear subgraph patterns described in WL-subtrees and the usage-based representations learnt of the subgraphs within the graphs leading to similar graphs being closer to each other, and dissimilar graphs more distant, allowing the C-SVM to compute more effective hyper planes which maximise the separation of graphs of different classes. The improvement over Graph2Vec as well as Deep Graph Kernels and AWE is based on the consideration of vocabulary pruning which overcomes the indirect issues of diagonal dominance (Yanardag & Vishwanathan, 2015) discussed in section 4.2.1. The heuristic can also be introduced to both of their systems. The unsupervised system can be shown to perform competitively against supervised GNN methods such as PATCHY-SAN and GIN, however we see its limitations in the unlabelled graph case in the next section.

The PTC dataset is a particularly challenging dataset for all methods. Despite being a relatively small dataset with an average graph size of 25.5 nodes, there are 19 distinct labels allowing a large number of different subgraph patterns which may have the same "shape" but are different structures as they have different node labels. This can make the attraction between graphs of the same class difficult as they may be exhibiting forms of diagonal dominance which cannot be alleviated by the encoding process of the embedding method particularly over the small number of graphs in the dataset. NCI1 and NCI109 in comparison have many more graphs in their dataset.

## 5.3 RESULTS: UNLABELLED GRAPHS

The social network datasets of Yanardag & Vishwanathan (2015), distinguish themselves from the previous set of graphs in 3 ways. Firstly, the graphs within the dataset are unlabelled hence the proposed system labels the nodes within the graphs by their degree and cannot be utilised in the

Table 2: Mean classification accuracy and standard deviation on unlabelled graph classification, italicised results indicate an unsupervised model for learning representations or kernels.

| Datasets | REDDIT-BINARY | REDDIT-5K | REDDIT-12K |
|---|---|---|---|
| *Graphlet Kernel* | *77.34 ± 0.18* | *41.01 ± 0.17* | *31.82 ± 0.08* |
| *Deep GK* | *78.04 ± 0.39* | *41.27 ± 0.18* | *32.22 ± 0.10* |
| *AWE (Data Driven)* | *87.89 ± 2.53* | *50.46 ± 1.91* | *39.20 ± 2.09* |
| PATCHY-SAN (K=10) | 86.30 ± 1.58 | 49.10 ± 0.70 | 41.32 ± 0.42 |
| 2D-CNN | 89.12 ± 1.70 | 52.11 ± 2.24 | 48.13 ± 1.47 |
| SUM-MLP (GIN-0) | **92.40 ± 2.50** | **57.50 ± 1.50** | - |
| SUM-MLP (GIN-$\epsilon$) | 92.2 ± 2.30 | 57.00 ± 1.70 | - |
| *G2DR + SVM* | *82.32 ± 1.21* | *52.82 ± 0.52* | *41.02 ± 0.91* |

unmodified Graph2Vec system. Secondly, the graphs are considerably larger than those seen in the classic benchmarks, having a direct effect on the complexity and number of substructures within, which in turn have a significant effect on the vocabulary size used in the neural language model. Finally, for REDDIT-5K and REDDIT-12K, is a multi-class classification task as opposed to the previous binary classification tasks. As a result, several of the previous methods are unsuitable for this task, and we have included another system, 2D-CNN (Tixier et al., 2019), which represents graphs as multi-channel image-like structures so that they can be fed into image convolutional neural networks. This system is another supervised model alongside PATCHY-SAN, GIN-0 and GIN-$\epsilon$.

As seen in table 2, G2DR performs better than the GK kernel and the Deep Graph Kernel variant Deep GK as expected. However, it fails to achieve the performances of supervised methods such as 2D-CNN and GIN as the specificity and sheer number, of unique subgraph patterns makes limitations of Graph2Vec apparent. A quick investigation showed that the lower performance could partly be attributed to the extremely large vocabulary of subtrees that is produced from the combination of very large complex graphs and a high number of data points in the dataset. This manifests into two related problems in the system. Foremost, whilst Graph2Vec reduces the effect of diagonal dominance by exploiting the distributional hypothesis and dimensionality reduction via the skipgram model, the sheer number of infrequently occuring subgraphs can still inadvertently cause diagonal dominance. Vocabulary pruning helps slightly, and can be seen push the performance slightly beyond AWE in the REDDIT5/12K problem set. AWE performs better than G2DR in the REDDIT BINARY dataset, as seen in graph kernel method work Vishwanathan et al. (2010) different substructures work better in different cases. This can also be seen in the labelled case amongst the deep graph kernels. Nonetheless the relatively high results of AWE and G2DR against supervised GNN methods, makes unsupervised distributed representation learning an interesting option for learning vector representations of graphs.

## 6 CONCLUSION

We have presented a general framework for constructing models which can learn distributed representations of graphs, that can be further generalised to other discrete structures under the R-Convolution kernel framework. This combines the theoretic frameworks of Deep Graph Kernels and advancements under Graph2Vec exemplified by our own instance, G2DR, which builds upon both previous works. G2DR is a scalable system which can learn task agnostic distributed representations of unlabelled/labelled and directed/undirected graphs of arbitrary size in a data driven manner. The suitability of this unsupervised approach is validated against graph kernels and supervised neural approaches in downstream graph classification tasks, where G2DR displays strong state of the art results despite its unsupervised nature and use of an off-the-shelf SVM. This enables the application of G2DR as a tool to numerous real world questions on graphs in different research domains. Furthermore, this suggests that the distributional perspective of graphs as compositions of discrete substructures is an useful inductive bias for building smooth vector representations of graphs along other research in hierarchical graph coarsening and substructure pooling.

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

## A    SUPPLEMENTARY MATERIAL: FIGURES AND TABLES

| Datasets | Size | Classes | Avg. Nodes | Distinct Labels |
|---|---|---|---|---|
| MUTAG | 188 | 2 | 17.93 | 7 |
| PTC | 344 | 2 | 25.5 | 19 |
| PROTEINS | 1113 | 2 | 39.06 | 3 |
| NCI1 | 4110 | 2 | 29.87 | 37 |
| NCI109 | 4127 | 2 | 29.68 | 38 |
| REDDIT-BINARY | 2000 | 2 | 429.61 | unlabelled |
| REDDIT-MULTI-5K | 4999 | 5 | 508.52 | unlabelled |
| REDDIT-MULTI-12K | 11929 | 11 | 391.41 | unlabelled |

Table 3: Properties of the benchmark graph classification datasets.

