# OpenReview forum: "How can we generalise learning distributed representations of graphs?"
_ICLR.cc/2020/Conference — Reject_

### Official Review · AnonReviewer2 · 2019-10-23
**Official Blind Review #2**

**Rating:** 6

**Review:**

This paper proposes a framework for learning distributional representations of graphs in the following way: First, each graph is represented as a collection of subtree patterns. Second, the neural language model of doc2vec is applied to these collections of patterns to learn graph embeddings. These embeddings are then exploited in downstream analyses such as classification. Overall, the idea of formulating graph representation learning as a language model is interesting. The experiments show that it perform better than kernel methods. I have the following major comments:

1. The main issue with this method is the computational complexity due to exponential growth of vocabulary of subtree patterns size for large graphs. Particularly , for experiments with unlabeled graphs, the performance is significantly worse than CNN based models. How would the performance be on unlabeled small graphs? For example, have you verified the performance on small graphs of section 4.2 when labels are ignored? (downstream clustering task)

2. The neural language models rely on the concept of context in documents. How the concept of context defined for subtree patterns extracted by Weisfeiler-Lehman algorithm?

3. The issue of diagonal dominance should be clarified. How does the pruning tackles this issue?



**Experience Assessment:**

I have published one or two papers in this area.

**Review Assessment: Checking Correctness Of Derivations And Theory:**

I assessed the sensibility of the derivations and theory.

**Review Assessment: Checking Correctness Of Experiments:**

I carefully checked the experiments.

**Review Assessment: Thoroughness In Paper Reading:**

I read the paper thoroughly.

---

> ### Author Response · Authors · 2019-11-07
> **Response for Blind Review #2**
>
> Thank you very much for reading this work and providing feedback. We will attempt to address each point individually.
>
> “1. The main issue with this method is the computational complexity due to exponential growth of vocabulary of subtree patterns size for large graphs. Particularly , for experiments with unlabeled graphs, the performance is significantly worse than CNN based models. How would the performance be on unlabeled small graphs? For example, have you verified the performance on small graphs of section 4.2 when labels are ignored? (downstream clustering task)”
>
> Indeed the computational complexity of this approach is high in the embedding learning stage due to the exponential growth of the subtree patterns extracted as the graphs get larger and more heterogeneous in terms of node labels. However we believe it is nonetheless interesting to look at alternative inductive biases (such as a distributive one, with various definitions of context) to learn representations of graphs. We believe intelligent definitions of “context” or vocabulary pruning can help significantly in this regard.
>
> We have not tried applying this to small unlabelled graphs. If time permits this will be in the revision (within an appendix) with the labeled datasets such as Mutag. Thank you for this suggestion.
>
> “  2. The neural language models rely on the concept of context in documents. How the concept of context defined for subtree patterns extracted by Weisfeiler-Lehman algorithm?”
>
> Yes defining the context is very important for learning useful distributive representations, and there are many different ways this can be done in natural language processing. For learning whole graph representations the context for a graph was its induced subtree patterns (which are extracted using the WL algorithm).
>
> “3. The issue of diagonal dominance should be clarified. How does the pruning tackles this issue?”
>
> We will attempt to describe the issue of diagonal dominance more concretely in the revision. Essentially diagonal dominance is related to the explosive increase of unique induced subgraph patterns when building our vocabularies. An example of this can be seen for our work, using the WL relabeling algorithm for the NCI1 dataset on the first iteration there are 267 subtrees, on the second there are 4033, and in the third iteration 22923 subtrees patterns within the graphs of NCI1. Consequently as the number of features (vocabulary size) grows, we run into the sparsity problem, where only a few substructures will be common across the graphs. This leads to the phenomenon known as diagonal dominance, where graphs become more similar to themselves but more distant from other graphs in the dataset. The naive pruning directly tackles this approach by removing dimensions along vocabulary instances that only appear a few times. Smarter ways of reducing this effect would lead to better distributed representations as we lightly touch upon in the discussion of the final results. We will try to make this more apparent in the revision. Thank you for this comment.
>
> We thank the reviewer for reading our work and the constructive feedback, we will work to integrate some of the comments into our revision.

---

### Official Review · AnonReviewer1 · 2019-10-23
**Official Blind Review #1**

**Rating:** 1

**Review:**

Strength:
-- The paper is well written and easy to follow
--  Learning the unsupervised graph representation learning is a very important problem
-- The proposed approach seems effective on some data sets.

Weakness:
-- The novelty of the proposed approach is very marginal
-- The experiments are very weak.

This paper studied unsupervised graph representation learning. The authors combined the techniques for Deep Graph Kernels and Graph2Vec, which essential extract substructures as words and the whole graph as documents and use doc2vec for learning the representations of both graphs and substructures. Experimental results on a few data sets prove the effectiveness of the proposed approach.

Overall, the paper is well written and easy to follow. Learning unsupervised graph representation learning is a very important problem, especially for predicting the chemical properties of molecular structures. However, the novelty of the proposed method is very marginal. Comparing to the Deep Graph kernel methods, the authors simply changed from the word2vec style methods to doc2vec style methods. The paper could be better fit to a more applied conference. Moreover,  I have some concerns on the experiments.
(1) The data sets used in this paper are too small. For unsupervised pretraining methods, much larger data sets are expected.

(2) The results in Table 1 are really weird. Why do the performance of your method have a much lower standard deviation? It is really hard to believe the proposed methods have much stable performance compare to other methods.  Can you explain this?

**Experience Assessment:**

I have published in this field for several years.

**Review Assessment: Checking Correctness Of Derivations And Theory:**

I carefully checked the derivations and theory.

**Review Assessment: Checking Correctness Of Experiments:**

I carefully checked the experiments.

**Review Assessment: Thoroughness In Paper Reading:**

I read the paper thoroughly.

---

> ### Author Response · Authors · 2019-11-07
> **Response for Blind Review #1**
>
> First of all thank you very much for your review of this work, we will attempt to address some of the comments and questions individually
>
> “This paper studied unsupervised graph representation learning. The authors combined the techniques for Deep Graph Kernels and Graph2Vec, which essential extract substructures as words and the whole graph as documents and use doc2vec for learning the representations of both graphs and substructures.”
>
> You are correct in this summary, however we may have not adequately stressed that the approach more generally highlights that graphs may be represented distributively via its internal substructure patterns (such as walks, nodes, induced subgraphs, subtrees, etc. as highlighted in Deep Graph Kernels [DGK]) as context. This allows a variety of embedding methods which exploit the distributive hypothesis to be applied on the graph-subpattern context pairs to learn vector representations of graphs (skipgram, cbow, GLOVE, pmi) etc. The revision will try to make this distinction clearer in the introduction. The intended contribution is an acknowledgement of the observation that many kernels fall under the R-Convolution framework (Haussler 1999) in DGK (Yanardag and Vishwanathan 2015); and generalisation beyond building representation with edit distance matrices and word2vec towards all embedding methods which exploit the distributive hypothesis.
>
> The 2nd half of this work presents G2DR (which is a straight-forward extension of the Graph2Vec model) to exemplify an instance of this framework using a decomposition of graphs to subtrees using the WL algorithm and building distributed representations with a skipgram model. This is just one possible instance of the approach above, and we chose to extend Graph2Vec (could have been called Graph2Vec2 but felt G2DR was more appropriate whilst acknowledging the previous work) as it could be modified to be utilised on a wider set of graph types.
>
> “However, the novelty of the proposed method is very marginal. Comparing to the Deep Graph kernel methods, the authors simply changed from the word2vec style methods to doc2vec style methods.“
>
> You are correct that Graph2Vec extends deep graph kernels through use of a WL subtree contexts followed by a skipgram architecture posed in the form of doc2vec. The contribution of G2DR is to modify the implementation the WL subtree decomposition with that in WL Kernel (Shervashidze et al, 2011) and take on their suggestion of relabeling unlabelled graphs by degree to allow building representations of unlabelled graphs (REDDIT graphs, for example). Furthermore we also attempt to lessen the problem of diagonal dominance in DGK/Graph2Vec by pruning the vocabulary of context subgraph patterns to improve downstream classification performance.
>
> Onto some of the questions:
> “(1) The data sets used in this paper are too small. For unsupervised pretraining methods, much larger data sets are expected. “
> Indeed it is difficult to find good large/public/popular datasets for comparative analysis with related works. As reported in the work we have sourced our datasets from https://ls11-www.cs.tu-dortmund.de/staff/morris/graphkerneldatasets (Kersting et al, 2016) around 2018. As on the list the REDDIT datasets are still the (2nd?) largest in this list and are regularly used in related literature which motivates its use in our work (as well as the fact it has unlabelled nodes).
>
> “(2) The results in Table 1 are really weird. Why do the performance of your method have a much lower standard deviation? It is really hard to believe the proposed methods have much stable performance compare to other methods.  Can you explain this?”
>
> Thank you for this observation! You are the only one that noticed this in the results table. The authors sincerely apologise for this mistake, the presented standard deviation comes from the 10 Fold SVM CV on the same embeddings output by G2DR through an old development file that reused pretrained embeddings for experiments. The revision will present the standard deviation of 10 Fold SVM CV being run on 10 trained embedding outputs of G2DR. This should give a more realistic picture of expected performance on the benchmarks.
>
> We hope that this clarifies some points and thank the reviewer for constructive feedback, and would be happy to discuss more.

---

### Official Review · AnonReviewer3 · 2019-10-23
**Official Blind Review #3**

**Rating:** 3

**Review:**

The paper presents an unsupervised method for graph embedding.

Despite having good experimental results, the paper is not of the quality to be accepted to the conference yet. The approach is rather a mix of previous works and hence not novel.

In particular, the algorithm for WL decomposition is almost fully taken from the original paper with a slight modification. Advantage of using it for unlabeled data is poorly motivated as unlabeled graphs can easily take statistics such as degree as the node labels, which was shown well in practice.

Modified PV-DBOW is in fact the same algorithm as the original CBOW model but applied to different context. It has been used in many papers, including Deep GK, graph2vec, anonymous walks.

Also, the Figure 1. is taken from the original paper of WL kernel. The algorithms 1 and 2 are taken from the original papers with slight modifications.

There is no discussion of [1], which uses CBOW framework, has theoretical properties, and produces good results in experiments. There is no comparison with GNN models such as [2].

I would be more interested to see explanation of the obtained results for each particular dataset (e.g. why MUTAG has 92% accuracy and PTC 67%); what so different about dataset and whether we reached a limit on most commonly used datasets.

[1] Anonymous Walk Embeddings? ICML 2018, Ivanov et. al.
[2] How Powerful are Graph Neural Networks? ICLR 2019, Xu et. al.

**Experience Assessment:**

I have published one or two papers in this area.

**Review Assessment: Checking Correctness Of Derivations And Theory:**

N/A

**Review Assessment: Checking Correctness Of Experiments:**

I carefully checked the experiments.

**Review Assessment: Thoroughness In Paper Reading:**

N/A

---

> ### Author Response · Authors · 2019-11-07
> **Response for blind review #3 Part 1**
>
> First of all thank you very much for your review of this work, we will attempt to address some of the comments and questions below.
>
> “Despite having good experimental results, the paper is not of the quality to be accepted to the conference yet. The approach is rather a mix of previous works and hence not novel.”
> And
> “In particular, the algorithm for WL decomposition is almost fully taken from the original paper with a slight modification... “
>
> This paper relies on previous models such as Deep Graph Kernels and Graph2Vec to extract and explicitly specify a general pipeline for building models capable of learning distributed representations of graphs.  The pipeline is based on two parts: the decompositions of graphs into substructures (walks, subtrees, nodes, etc) and the learning distributed representations using such substructures with different definitions of context and associated embedding methods (word2vec, GLoVe, etc.).
>
> The second half of the write-up focuses on G2DR (explicitly stated as an extension of Graph2Vec) as an instance of this pipeline described above. G2DR is a straightforward extension of the Graph2Vec to more graph types (unlabelled graphs) through adoption of Shervashidze et al’s WL algorithm to find subtree patterns, we have put it in this work with minor modification for notation because otherwise it wouldn’t be the same WL algorithm. We believe in keeping the algorithm in the paper as it aids description of the specific implementation used and is correctly acknowledged as being the Shervashidze WL algorithm within the paper (section 3.1.1). We are afraid that simply pointing to the Shervashidze et al’s exact presentation would detract from the reading and flow of the paper as different notation is used.
>
> To summarise we can garner two contributions here:
> Specification of a general pipeline for building models capable of learning distributed representations of graphs.
> An extended version of Graph2Vec, called G2DR which is applicable to unlabelled graphs and is also more amenable to diagonal dominance through pruning of the subgraph vocabularies. This makes it perform better on larger graphs/datasets.
>
> ”Advantage of using it for unlabeled data is poorly motivated as unlabeled graphs can easily take statistics such as degree as the node labels, which was shown well in practice.”
>
> We explicitly state our use of Shervashize et al’s suggestion to label unlabelled nodes initially by their degree, otherwise the WL algorithm cannot be run for the unlabelled graphs such as the Reddit datasets. The contribution here is the application of this suggestion within another existing algorithm (Graph2Vec) to expand its applicability to more graph types and improve the performance of the GetSubgraph() (which is their rendition of the subtree decomposition algorithm) algorithm stated in Graph2Vec.
>
> Once the unlabelled nodes are labelled by their degree, the motivation of using the WL algorithm falls upon motivating the usage of the rooted subtree patterns extracted. We touch upon this section 3.1.1 and is potentially better covered in the WL Kernel and Graph2Vec works. Essentially the motivation is that they are higher order substructures (than nodes), non-linear around definition of the neighbourhood around a node (as compared to a random walk), and the exhaustive nature of decomposition for subtree patterns for every node in the graph is useful to characterise all the patterns (subtree patterns) within a given graph. Another pragmatic motivation is that the WL Kernel has been shown to work well in graph classification tasks. We will try to make these motivations more clear in the paper, thank you for this comment and suggestion.
>
> “Modified PV-DBOW is in fact the same algorithm as the original CBOW model but applied to different context. It has been used in many papers, including Deep GK, graph2vec, anonymous walks. “
>
> Yes you are completely correct! We explicitly say that we are using the embedding method from Graph2Vec (hence the name of the algorithm also being TrainGraph2Vec). We kept the misleading Doc2Vec analogies used in Graph2Vec as it aided exposition of how one can think of a graph as composition of substructures, like documents being compositions of words. As the contexts of the graphs are defined as the subtree patterns within it, it is actually more similar to training a word2vec model as you mention. To make this clear we will change the title of this section in the revision. Thank you for this comment.

---

> > ### Author Response · Authors · 2019-11-07
> > **Response for Blind Review #3 Part 2 (of 2)**
> >
> > “Also, the Figure 1. is taken from the original paper of WL kernel. The algorithms 1 and 2 are taken from the original papers with slight modifications. “
> >
> > As previously we explicitly say we use the algorithms from their respective papers with acknowledgement to aid description of the G2DR and Graph2Vec approaches with notational changes for consistency in the explanations.
> >
> > -For algorithm 1 “WL-Relabel” in section 3.1.1. <<... This is achieved as a byproduct of WL test’s node relabeling (Shervashidze et al,. 2011) and is fully described in algorithm 1 ...”>>
> >
> > -For algorithm 2 “Train-Graph2Vec” in section 3.1.3 <<... We follow Graph2Vec (Narayanan et al., 2017) and use a PV-DBOW… as outlined in algorithm 2>>
> >
> > We think presenting the algorithms helps the reader refer to details within the paper itself to get more exposition if they wish to do so with notation that is consistent within this work. However together with the comment on the PVDBOW name being an ill-suited name for the method we may remove algorithm 2 and replace it with the objective function of the word2vec/graph2vec algorithm to save space and address other points. Similarly for the figure 1 which was used for explanation purpose (it actually has an additional node number 5 in comparison to the figure in the WL Kernel, so the extracted subtree is accordingly different as well), we may remove this in favour of addressing some of the other points in the reviews, or make a showcase of the subtree extraction on a more obviously different graph. Thank you for the comment and we will revise the document as necessary.
> >
> > “There is no discussion of [1], which uses CBOW framework, has theoretical properties, and produces good results in experiments. There is no comparison with GNN models such as [2]. “
> >
> > Thank you for introducing us to [1] (AWE). We have simply not come across this work during the time working on this project. This is a very nice paper with clear parallels to this work as it has to DGK and Graph2Vec as well. In fact the style of this work is very similar to Graph2Vec with the usage of anonymous walks instead of subtree patterns as input into context based language models. In one sense the AWE is another method that can fall under the framework described in the introduction and section 3 alongside DGK, Graph2Vec, G2DR. This is very neat and thank you for pointing this out, we will try to include it in the revision and results tables.
> >
> > Thank you for pointing out [2]. Our original comparison for GNN/convolution based methods was to compare between DiffPool and PATCHYSAN as described in the paper. Because DiffPool had only published results for one of the datasets within our selection without standard deviations we did not include this, in favour of PATCHY-SAN which did cover all the datasets with appropriate standard deviations. The GIN in [2] seems to have results for almost all of the datasets so we will include it in the results table of the revision. To help clarify relations with the GNN based methods we will retitle section 2.2 as “Deep learning approaches: GNNs and convolutional approaches”.
> >
> > “I would be more interested to see explanation of the obtained results for each particular dataset (e.g. why MUTAG has 92% accuracy and PTC 67%); what so different about dataset and whether we reached a limit on most commonly used datasets. “
> >
> > Yes thank you, this is an interesting question on what defines the difficulty of the classification tasks based on the properties of the dataset. From the point of view of building distributed representations we think an interesting way to look at it would be the characterisation of the substructure pattern distributions for graphs of different classifications. In PTC there may be clear overlap between the distributions which makes it hard to make representations that are easy to seperate downstream. If time permits within the revision period we will either answer directly on a comment or within an appendix section.
> >
> > We thank the reviewer for reading our work and the constructive feedback. We will work to incorporate tips from the comments into the revision.

---

> > > ### Comment · AnonReviewer3 · 2019-11-14
> > > **Still I am not convinced**
> > >
> > > - as we already discussed, the approach is the mix of several existing approaches
> > > - one of the main differences of the proposed approach is that the authors use some other method (which is already known) to construct a vocabulary of subtree patterns for large graphs
> > > - the computational complexity of the proposed approach is high due to high computational complexity of constructing the vocabulary. Is it worth using this approach due to its high computational cost?
> > > - the experimental section is rather weak. Since the approach is based on using different building blocks, it is necessary to know which of building blocks is the most important and provides the most contribution to increase in accuracy. Is it due to the new vocabulary? Or doc2vec? or what?
> > > - as a results still I am not convinced in the provided comments.

---

> > > > ### Author Response · Authors · 2019-11-14
> > > > **Thank you for your comment**
> > > >
> > > > Thank you for your comment
> > > >
> > > > "- as we already discussed, the approach is the mix of several existing approaches"
> > > >
> > > > As in the first response, the paper does not describe is not a single approach or single model exclusively, we are highlighting a common framework/workflow utilised by other models which learn distributed representations of graphs (deep graph kernels, graph2vec, anonymous walk embeddings can all be described within this framework). The second portion presents an extended version of Graph2Vec described using this framework as an example, we call this G2DR for easier reference. We will revise the abstract to make this more clear from the start.
> > > >
> > > > "-one of the main differences of the proposed approach is that the authors use some other method (which is already known) to construct a vocabulary of subtree patterns for large graphs"
> > > >
> > > > Yes one extension to the current implementation of Graph2Vec is to use the more general WL node relabeling algorithm described by Shervashidze et al. We don't believe the use of a good algorithm described by a well known paper in graph kernels is a bad thing.
> > > >
> > > > "- the computational complexity of the proposed approach is high due to high computational complexity of constructing the vocabulary. Is it worth using this approach due to its high computational cost?"
> > > >
> > > > The construction of the substructure vocabulary is O(|G|dm) where |G| is the number of graphs in the dataset, d is the highest degree of subtree pattern we wish to extract, and m is the highest number of edges in a graph within the dataset. An advantage of using the distributed approach is clear associations that can be drawn between graphs that are deemed similar as we can expect and actually present the different subgraph patterns as they are recorded as contexts. This can be useful subsequent analysis on the motifs present in different classes of graphs.
> > > >
> > > > "- the experimental section is rather weak. Since the approach is based on using different building blocks, it is necessary to know which of building blocks is the most important and provides the most contribution to increase in accuracy. Is it due to the new vocabulary? Or doc2vec? or what?"
> > > >
> > > > The presented results are a result of the pruned vocabulary. This is the only operation that differs from Graph2Vec that directly affects the learning model (skipgram).
> > > >
> > > > The other "building blocks" such as using Shershavidze et al's WL-relabeling algorithm and suggestion on labeling nodes by degree allows application of the model to learn distributed representations of unlabelled graphs; hence the results on the Reddit datasets. Otherwise the WL algorithm produces exactly the same subtrees as the WL algorithm described in Graph2Vec in the labeled case; which then get pruned based on their frequency.
> > > >
> > > > Once again thank you for your comment. Hopefully this addresses some of the new questions.

---

> > > > > ### Comment · AnonReviewer3 · 2019-11-15
> > > > > **Building blocks**
> > > > >
> > > > > In general I am ok with comments of the authors.
> > > > >
> > > > > However, I think that it is rather obvious (taking into account so many papers on this topic) a general structure of such type of algorithms, what kind of building blocks we should use. I think that the main remaining issue here is to improve some of the blocks and prove under which conditions those modifications can have some impact on theoretical properties, e.g. ability to solve graph isomorphism.
> > > > >
> > > > > Therefore, I do not think I can increase my grade significantly.

---

### Author Response · Authors · 2019-11-15
**Uploaded Revision**

We would like to thank all of the reviewers for reading our work and providing feedback to improve our work and correct mistakes.

We have taken these into consideration and uploaded a revision. On top of including as many of the pointers and promised revisions as possible, we have changed parts of the presentation to be clearer on the different contributions.

Main points of revision:
- We have updated the abstract to be clearer about the specific contributions of this work. (unfortunately we cannot update the abstract on this webpage.)

- We have included suggested related work from Reviewer #3 and made comments within the paper, as well as including their results on our selection of benchmark datasets.

- We have expanded on our heuristic to prune the subgraph pattern vocabulary to handle the indirect influence of diagonal dominance as suggested by Reviewer #2 within Section 4.2.1. We point out how this helps the skipgram model learn a more useful representations (for the downstream graph classification task).

- We have attempted to better separate the description of the framework for building models which can learn distributed representations of graphs, and the presentation of an extended version of Graph2Vec, described using this framework.

- Changes to the section titles and seperation of some previous subsections into their own to reflect above point, and make the content clearer.

- Removal of figure 1 and Algorithm 2, replaced with equation of the objective function to be optimised in the learning phase to save space and include revisions promised elsewhere.

- Updated the results tables, as we previously presented the standard deviation of the downstream SVM instead of multiple iterations of the entire system as pointed out by Reviewer #1. We have also included the suggested models of Reviewer #3.

- We have updated sections 5.2 and 5.3 to give a better discussion on the results, comparison with other approaches, and a comment on the PTC dataset and the challenge it poses to graph classification systems.

Once again we would like to thank all who have read our work, and provided pointers for improvement in the revision and future.

---

### Decision · Program_Chairs · 2019-12-19

**Decision:**

Reject

**Comment:**

The paper proposed a general framework to construct unsupervised models for representation learning of discrete structures. The reviewers feel that the approach is taken directly from graph kernels, and the novelty is not high enough.